# Modeling Storm Surge Attenuation by an Integrated Nature-Based and Engineered Flood Defense System in the Scheldt Estuary (Belgium)

**Sven Smolders** [1,2,*] **, Maria João Teles** [3]**, Agnès Leroy** [3]**, Tatiana Maximova** [1]**, Patrick Meire** [2] **and Stijn Temmerman** [2]

1    Flanders Hydraulics Research, Department of Mobility and Public Works, Flemish Government, 2660 Antwerp, Belgium; tatiana.maximova@mow.vlaanderen.be

2    Ecosystem management research group, Department of Biology, University of Antwerp, Universiteitsplein 1c, 2610 Antwerp, Belgium; patrick.meire@uantwerpen.be (P.M.); stijn.temmerman@uantwerpen.be (S.T.)

3    EDF R&D/LNHE & Université Paris-Est, Saint-Venant Hydraulics Laboratory (ENPC, EDF R&D, CEREMA), 78400 Chatou, France; maria.teles@edf.fr (M.J.T.); agnes.leroy@edf.fr (A.L.)

*    Correspondence: svensmolders@gmail.com or sven.smolders@mow.vlaanderen.be

**Abstract:** There is increasing interest in the use of nature-based approaches for mitigation of storm surges along coasts, deltas, and estuaries. However, very few studies have quantified the effectiveness of storm surge height reduction by a real-existing, estuarine-scale, nature-based, and engineered flood defense system, under specific storm surge conditions. Here, we present data and modelling results from a specific storm surge in the Scheldt estuary (Belgium), where a hybrid flood defense system is implemented, consisting of flood control areas, of which some are restored into tidal marsh ecosystems, by use of culvert constructions that allow daily reduced tidal in- and outflow. We present a hindcast simulation of the storm surge of 6 December 2013, using a TELEMAC-3D model of the Scheldt estuary, and model scenarios showing that the hybrid flood defense system resulted in a storm surge height reduction of up to half a meter in the estuary. An important aspect of the work was the implementation of model formulations for calculating flow through culverts of restored marshes. The latter was validated comparing simulated and measured discharges through a physical scale model of a culvert, and through a real-scale culvert of an existing restored marsh during the storm surge.

**Keywords:** culvert flow; TELEMAC-MASCARET; nature-based flood defense; storm surge attenuation

## 1. Introduction

Storm surges, caused by low atmospheric pressure and strong winds, are the main mechanism responsible for coastal flooding around the world [1]. Storm surge flood disasters like hurricane Katrina in New Orleans (2005), hurricane Sandy in New York (2012), typhoon Haiyan in the central Philippines (2013), and hurricane Dorian in The Bahamas (2019) are some recent examples of the devastating effects of coastal flooding. Coasts, deltas, and estuaries are prone to increasing storm surge flood risks [2–7] due to several aspects of global change, including sea level rise, increasing storm activity, growing coastal populations, and land subsidence in many deltaic and estuarine settings. Today, an estimated 600 million people live in flood-prone coastal areas [8] and by 2050 this is expected to increase to a billion people [9].

Hence, there is a globally increasing demand for developing and implementing innovative and effective flood defense programs. In addition to traditional engineering approaches, such as dikes and flood barriers, there is an increasing interest in so-called nature-based and hybrid (i.e., combined nature-based and engineered) approaches for mitigating storm surge flood risks [10–13]. These include studies identifying the impact of naturally existing intertidal ecosystems (marshes, mangroves) on attenuation of storm surge water levels, when storm surges propagate through such intertidal wetlands or through estuarine or deltaic channels with adjacent wetlands [14–19]. Hybrid approaches include the combination of natural or (re-)constructed wetlands with engineered flood defense structures [10,11]. However, very few studies have quantified the effectiveness of storm surge reduction by a real-existing, estuarine-scale, nature-based or hybrid flood defense scheme, under specific storm surge conditions.

Here, we present data and modelling results assessing the effectiveness of a recently implemented, estuarine-scale flood defense program that combines a nature-based and engineering approach (the Sigmaplan, Scheldt estuary, Belgium) This estuary is adjacent to the North Sea, which is subjected to some of the highest storm surge levels in Europe and these levels are expected to increase in the future due to climate change projections (increase in significant wave height and storm surge residual [20]; increase in sea level and wind speed [21] and increase in surface wind speed [22]). Our study focusses on storm 'Xaver' that struck the North Sea region on 6 December 2013. In the Netherlands and Belgium, this storm was called 'Sinterklaas' storm after Saint Nicholas, which is locally celebrated on that day. This flood event was a superposition of a spring tide and a storm surge due to strong northwestern winds over the North Sea resulting in record-breaking sea levels for several locations along the eastern North Sea coast (e.g., [23–26]). The return period of this storm was 19 years at Vlissingen, i.e., the estuary mouth [27]. Flood damage was relatively limited because nowadays effective flood defense systems are in place. The latter is one of the reasons this storm was chosen for this study, because sufficient water level measurements were available, including water level measurements inside recently implemented flood plains.

On the one hand, there are strongly engineered flood defense systems, such as the Thames storm surge barrier in London [28], and the Delta plan in the Netherlands that includes closure dams and storm surge barriers in most estuaries in the Dutch Delta [29]. These hard engineering structures were built after the disastrous 1953 storm surge that killed about 2000 people along the North Sea coasts and estuaries.

On the other hand, in the Belgian Scheldt estuary, it was after a storm surge in 1976 that the government started with their own flood protection plan, called 'Sigmaplan', which evolved over the years into a combined nature-based and engineered flood defense system. The original plan contained three main objectives: (1) Heightening and strengthening the dikes; (2) building a storm surge barrier; (3) creating floodplains, called flood control areas (FCAs), to temporally store storm surge water. In 2005, the Sigmaplan was updated to include more nature-based elements and to obtain a more holistic estuarine management plan. This means that the Sigmaplan is no longer aiming only at flood prevention, but also at restoring the ecological values of the estuary, in combination with the improvement or conservation of other estuarine functions, like port accessibility or recreation [30]. Tidal marshes that were historically embanked are restored by new de-embankments and, even outside the flood-protecting dikes, wetlands are created to restore the ecological value of the Scheldt valley. The construction of the storm surge barrier was not realized. Instead, a hybrid nature-based and engineered defense system was implemented by creating more than 2500 ha of new floodplains (FCAs) [31]. During normal tidal conditions, these floodplains are separated from the estuary by dikes, which are designed as such that during storm surge conditions, the dikes are overtopped by the surge level, so that water flows into the floodplains, and the storm surge level in the estuary is reduced. These floodplains are surrounded by flood-protecting ring dikes to prevent flooding of the hinterland. This system of floodplains must protect the estuary against storm tides with a return period of 4000 years [31]. Some of these floodplains are restored into tidal marsh ecosystems by re-introduction of tides into historically embanked areas. This daily tidal exchange is realized by culverts through the dikes to

obtain a controlled reduced tide (CRT) that enables the development of tidal marsh ecosystems into the FCA [31–34]. Thereby, taking the benefits from nature development (e.g., [31–34]) into account, this hybrid flood defense program has a better cost–benefit ratio as compared to a traditional hard engineering approach only consisting of dikes and/or a storm surge barrier [31,35].

In this paper, we present a hydrodynamic modelling study (using TELEMAC-MASCARET software), aiming to assess the effectiveness of storm surge height reduction that is caused by the presence of constructed FCAs with and without CRT during the 2013 'Sinterklaas' storm in the Scheldt estuary (Belgium). In order to achieve this goal, the modelling study consists of the following steps. This paper starts from the development and implementation of equations in TELEMAC, which allow to model flow through culverts of FCAs with CRT in the Scheldt estuary [36]. The presented modelling approach is validated by comparing measured and modeled discharges and water levels in a detailed 3D model of one FCA with CRT. Next, discharges measured through a physical scale model of a CRT culvert are compared with modeled discharges. Finally, the new culvert flow implementation is used in a 3D hydrodynamic model of the whole Scheldt estuary to hindcast the 'Sinterklaas' storm and to show the impact of FCAs on the attenuation of the storm surge along the estuary.

## 2. Materials and Methods

### 2.1. Study Case

#### 2.1.1. The Scheldt Estuary

The Scheldt estuary is located in Western Europe in the Netherlands and Belgium (Figure 1a). The part of the estuary from the mouth until the Dutch/Belgian border (located at 67 km from the mouth, measured along the thalweg) is called Western Scheldt and is characterized by different ebb and flood channels and adjacent large intertidal flats and marshes. The tides of the Scheldt estuary are semi-diurnal. At the mouth, near Vlissingen (for location see Figure 1), the estuary is approximately 5 km wide and has an average tidal volume of 1.04 $Gm^3$ [37]. The part upstream the border until Merelbeke (located at 170 km from the mouth) is called Sea Scheldt and is characterized by a single channel bordered by much smaller intertidal flats and marshes. In Merelbeke, the tide is blocked by locks and a weir. For clarity, because water flows in both directions in an estuary, downstream depicts the direction towards the sea and upstream is directed towards the inflow of fresh water.

The Scheldt estuary is a hypersynchronous estuary, where tidal damping by friction is less than tidal amplification by upstream convergence. Therefore the tidal range amplifies, for mean spring and neap tides respectively, from 4.46 m and 2.97 m at the mouth to 5.93 m and 4.49 m near Hemiksem (located at km 104 from the estuary mouth, see Figure 1). Further upstream tidal damping by friction becomes dominant and this results in a mean tidal range of 2.24 m and 1.84 m for spring and neap tide respectively near Merelbeke (located at km 170 from the estuary mouth, see Figure 1). The total discharge of the Scheldt (on average 120 $m^3$/s) is very small compared to the tidal volume [32,38].

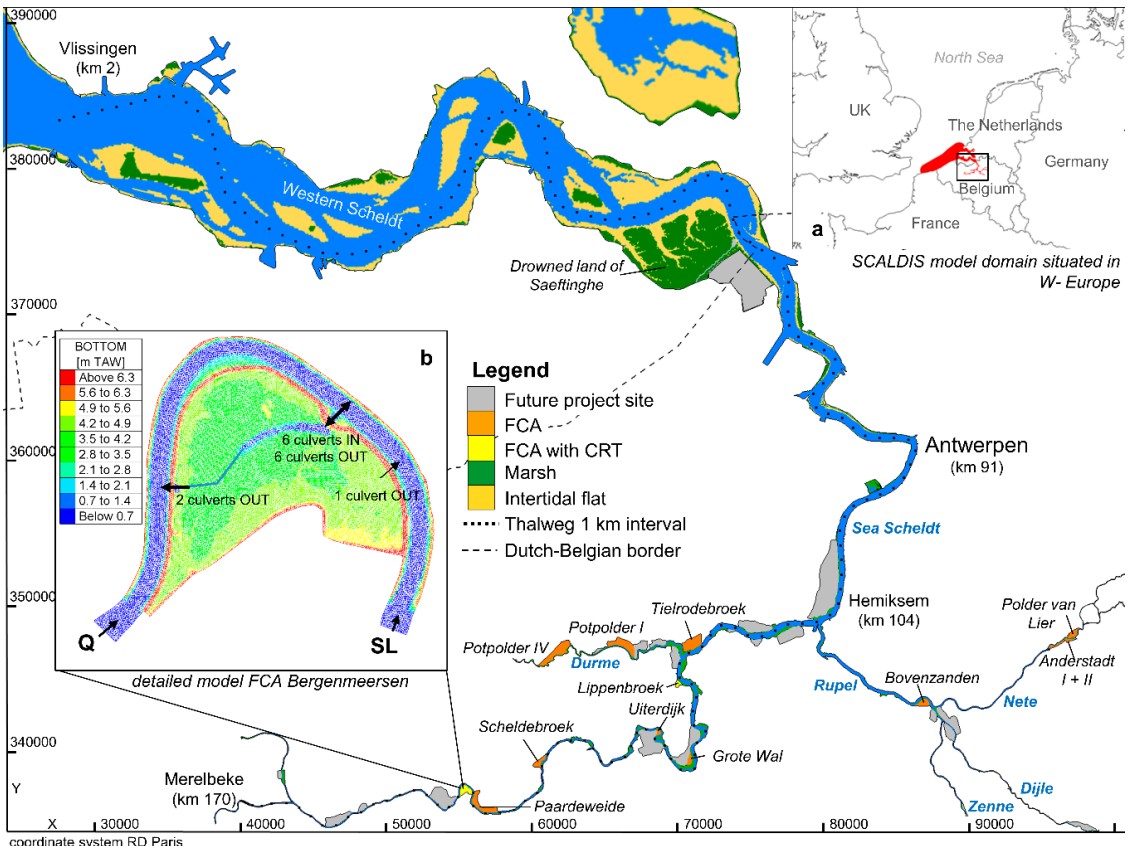

**Figure 1.** Map of the Scheldt estuary in 2013 (i.e., the year of the simulated storm surge, see text) with names of all active flood control areas (FCAs). (**a**) The SCALDIS model domain located in W-Europe; (**b**) model domain and bathymetry of the detailed model of FCA Bergenmeersen with Q indicating the upstream discharge input and SL indicating the downstream surface level boundary (TAW is the Belgian vertical reference plane where 0 m TAW corresponds to low water level at sea).

### 2.1.2. FCAs and CRT

Along the Scheldt estuary the polders are very low-lying lands compared to the water levels in the estuary. This is due to historic early dike building and thus the lack of sedimentation on these areas. They did not follow the increasing high water levels like the natural marshes within the estuary dikes [39]. The advantage for the FCA is that these low-lying polders provide extra storm water storage capacity. When a specific polder area is designated to become FCA, a ring dike, protecting the lands behind it from flooding, is first constructed around the polder. Then, the existing dike between the polder and the estuary is lowered to a specific chosen level. The level of this overflow dike and its length determine the start and amount of storm water that can enter the FCA. The level and length of the overflow dike are specific for each FCA and depend highly on the location along the estuary. When, after a storm, surge the water levels in the estuary are sufficiently low again (ebb tide), the storm water is drained from the FCA through outlet culverts built in the overflow dike. These outlet culverts are equipped with flap gates or one-way valves to prevent water entering the FCA each normal tide. When a storm surge is entering the estuary and the water level reaches above the level of the overflow dike, storm water enters the FCA and is stored there, effectively damping the tidal wave along the estuary and protecting areas upstream.

In some FCAs, the flood protection function is combined with estuarine nature restoration [31–34,39]. To do so, tidal water is introduced in the FCA through a simple design of high inlet culverts and low outlet culverts. To mimic the hydrology of tidal marshes inside the polder, a reduced tidal regime is necessary. Introducing the full tidal regime would otherwise flood the

entire polder, which is not beneficial for tidal wetland development [31,39,40]. This type of FCA is called FCA with CRT. The inflow culvert dimensions and bottom level will determine the tidal regime inside the FCA. A sketch of the FCA with CRT Bergenmeersen is shown in Figure 2. This figure shows a cross section of the FCA with the ring and overflow dike and the construction of the inflow culverts just above the outflow culverts. Other FCAs with CRT can have different types of constructions, for example where the in- and outflow culverts have a different location along the overflow dike [41]. The in- and outflow culverts are equipped with trash screens to prevent debris from clogging the culvert flow (indicated by number 4 in Figure 2). The inflow culvert can be closed by a down sliding valve (indicated by number 3 in Figure 2). In case of a storm surge, the inflow culverts can be closed to maintain a maximum storage capacity for the area to function as FCA. In case the tidal volume entering the FCA with CRT needs to be regulated a little to meet changing design criteria for the inside flood plain, wooden beams (indicated by number 2 in Figure 2) can be placed in front of the inflow culverts to delay the start of inflow.

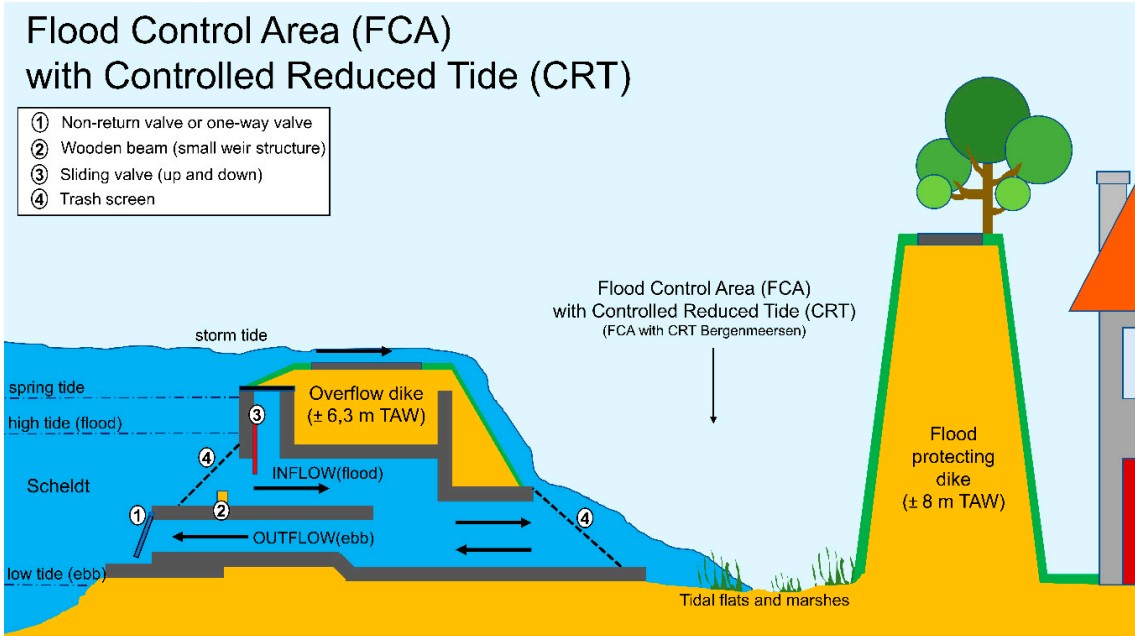

**Figure 2.** Cross section of flood control area (FCA) with controlled reduced tide (CRT) Bergenmeersen also showing the internal structure of the in- and outflow culverts. TAW is the Belgian vertical reference plane where 0 m TAW corresponds to low water level at sea.

### 2.2. TELEMAC-3D Model: SCALDIS

SCALDIS is a 3D hydrodynamic model of the Scheldt estuary developed at Flanders Hydraulics Research for various applications. It uses the software TELEMAC-3D, which is one of the finite element software modules of the open TELEMAC-MASCARET suite. In TELEMAC-3D, the three-dimensional Reynolds averaged Navier–Stokes equations are solved hydrostatically or non-hydrostatically [42]. The model domain includes the entire Scheldt estuary with tributaries as far as the tidal influence reaches, as well as the Belgian coastal zone, parts of the adjacent French and Dutch coastal zones, and the adjacent Eastern Scheldt estuary (see Figure 1). The domain has eight upstream boundaries, where boundary conditions are defined by daily averaged measured discharges. Downstream, water-level time series, extracted from a continental shelf model, are imposed on the seaward boundary. The horizontal mesh resolution varies from 500 m in the coastal zone to 120 m in the Western Scheldt, to 60 m in the Sea Scheldt further increasing upstream towards 3 m at the upstream boundaries. The horizontal grid contains 459,692 nodes. The vertical grid contains five layers following a sigma transformation (layers at 0% 12%, 30%, 60%, and 100% of the water depth). The model bathymetry and

intertidal topography is interpolated from multi-beam (with a resolution of $20 \times 20$ m for the Western Scheldt and $5 \times 5$ m for the Sea Scheldt) measurements and Lidar data (with a resolution of $5 \times 5$ m) respectively. The model represents the estuary as it was in 2013, which is the year of the simulated storm surge (see below). Salinity is an active tracer and density effects are taken into account. The mixing length model of Nezu and Nakagawa [43] is used for the vertical turbulence modelling. For the horizontal turbulence, the Smagorinsky model [44] is used. The Coriolis force is taken into account.

The model was calibrated by spatially varying the values of a Manning bottom friction coefficient. The model was calibrated extensively with measured water levels at 34 tide gauge stations, flow velocities and discharges over 23 ADCP transects, and 15-point measurements of flow velocities. For more detailed information, we refer to the calibration report [45].

The model domain includes all FCAs foreseen in the Sigmaplan (Figure 1), but in 2013, only 13 were already built. This means the planned but not yet realized FCAs were switched of in the model. Two of the functioning FCAs in 2013 are working with CRT: Bergenmeersen and Lippenbroek (Figure 1). The latter was the pilot area to test the CRT design principle [31,39,41]. Two FCAs are located along the Durme tributary; four FCAs are located upstream the Rupel tributary; and seven are located along the Scheldt. All locations are shown in Figure 1.

### 2.3. Model Implementation of Culvert Flow

### 2.3.1. Different Types of Flow

Bodhaine categorized the flow through a culvert into six types [46], and for each type, the discharge is calculated in a different way. The equations are deduced from the continuity and energy equations between the approach and exit (downstream) section of the culvert. The type of flow depends on whether the flow fills the whole culvert and whether it is controlled by the entrance or exit water level of the culvert. Table 1 summarizes the six discharge equations for each type of flow defined by Bodhaine and when to apply which flow type. With flow type 1, the critical depth occurs at the entrance of the culvert and the flow is supercritical inside. The culvert slope is greater than the critical slope and the culvert flows partially full. Because the culverts in the FCAs along the Scheldt are all horizontally constructed, this flow type will not be taken into account in the rest of this paper. Flow type 1 is also not implemented in TELEMAC. With flow type 2, the critical water depth is located at the exit of the culvert and the flow is subcritical inside. With flow type 3, there is no critical depth: The flow is tranquil in and outside the culvert. With flow type 4, the in- and outlet of the culvert are submerged: The flow fills the whole culvert. With flow type 5, the flow is supercritical at the entrance of the culvert: The flow fills the culvert only partially. With flow type 5, the discharge coefficient is in general lower than with the other types of flow. With flow type 6, the flow fills the whole culvert and the difference with flow type 5 depends on the culvert's geometry, i.e., the ratio of its length and height [46].

**Table 1.** Summary of the different flow types through a culvert by Bodhaine [46], how to calculate the discharge, and when to use which equation.

| Flow Type | Discharge Equation | Occurs When |
|:---:|:---:|:---:|
| 1 | $Q = C_D A_c \sqrt{2g\left(h_1 - z - h_c - h_{f12} + \alpha \frac{V_1^2}{2g}\right)}$ | $\frac{h_1-z}{D} < 1.5; \frac{h_4}{h_c} < 1.0; S_0 > S_c$ |
| 2 | $Q = C_D A_c \sqrt{2g * \left(h_1 - h_c - h_{f12} - h_{f23} + \alpha \frac{V_1^2}{2g}\right)}$ | $\frac{h_1-z}{D} < 1.5; \frac{h_4}{h_c} < 1.0; S_0 < S_c$ |
| 3 | $Q = C_D A_3 \sqrt{2g\left(h_1 - d_3 - h_{f12} - h_{f23} + \alpha \frac{V_1^2}{2g}\right)}$ | $\frac{h_1-z}{D} < 1.5; \frac{h_4}{D} \leq 1.0$ |
| 4 | $Q = C_D A_0 \sqrt{\frac{2g(h_1-h_4)}{1+29C_D^2 n^2 L/R^{4/3}}}$ | $\frac{h_1-z}{D} > 1.0; \frac{h_4}{D} > 1.0$ |
| 5 | $Q = C_D A_0 \sqrt{2g(h_1 - z)}$ | $\frac{h_1-z}{D} \geq 1.5; \frac{h_4}{D} \leq 1.0$ |
| 6 | $Q = C_D A_0 \sqrt{2g\left(h_1 - d_3 - h_{f23}\right)}$ | $\frac{h_1-z}{D} \geq 1.5; \frac{h_4}{D} \leq 1.0$ |

With: $Q$, the discharge through the culvert; $C_D$, the discharge coefficient; $A_c$, the flow area at the critical water depth; $g$, the gravitational constant; $h_1$, the upstream water depth; $z$, the elevation of the culvert entrance; $h_c$, the critical water depth; $h_{f12}$, the head loss due to friction from the approach section to the culvert entrance; $\alpha$, kinetic energy correction coefficient for the approach section; $V_1$, the average flow velocity at the approach section of the culvert; $h_{f23}$, the head loss due to friction inside the culvert; $A_3$, the flow area at the culvert outlet; $d_3$, water depth at the culvert outlet; $A_0$, flow area at the culvert entrance; $h_4$, the downstream water depth; $n$, the Manning friction coefficient; $L$, the length of the culvert; $R$, the hydraulic radius; $S_0$, the culvert slope; $S_c$, the critical slope.

### 2.3.2. Reformulation of the Equations for Model Implementation

Head loss coefficients are the topic of different studies often found through scale-model experiments. A number of authors have arrived at different values or empirical relationships for the head loss coefficients. Bodhaine suggests different values for the discharge coefficient ($C_D$) for each flow type, depending on a number of geometric properties of the culvert [46]. The discharge coefficients can vary from 0.39 to 0.98. A different approach is proposed by Carlier [47] using a non-dimensional coefficient, also referred to as a discharge coefficient that, for hydraulic structures made of only one culvert, can be written as follows:

$$\mu = \frac{1}{\sqrt{C_1 + C_2 + C_3}} \tag{1}$$

where $C_1$ is the head loss coefficient at the entrance of the culvert, $C_2$ is the head loss coefficient inside the culvert, and $C_3$ is the head loss coefficient at the exit of the culvert. Then, if the general expression for the discharge equation through a culvert proposed by Carlier [47]:

$$Q = \mu A \sqrt{2g\Delta H} \tag{2}$$

is compared with the formulae for the different flow types given by Bodhaine [46], the non-dimensional discharge coefficient ($\mu$) incorporates both the effect of the discharge coefficient, $C_D$, and the continuous and local head losses. $\Delta H$ is the head for each type of flow. The equations of Bodhaine in Table 1 can be rewritten according to the format proposed by Carlier [47] and can in this form be implemented in the TELEMAC code. The equations following Carlier [47] and the conditions of when to use which equation are summarized in Table 2.

**Table 2.** Summary of the discharge equations for the different flow types through a culvert following Carlier [47].

| Flow Type | Discharge Equation | Occurs When |
|:---:|:---:|:---:|
| 2 | $Q = \mu h_c W \sqrt{2g*(S_1-(z_2+h_c))}$ | $\frac{S_1-z_1}{D} < 1.5$; $S_2 - z_2 < h_c$ |
| 3 | $Q = \mu(S_2-z_2)W\sqrt{2g(S_1-S_2)}$ | $\frac{S_1-z_1}{D} < 1.5$; $\frac{S_2-z_2}{D} \leq 1.0$ |
| 4 | $Q = \mu DW\sqrt{2g(S_1-S_2)}$ | $\frac{S_1-z_1}{D} > 1.0$; $\frac{S_2-z_2}{D} > 1.0$ |
| 5 | $Q = \mu DW\sqrt{2gh_1}$ | $\frac{S_1-z_1}{D} \geq 1.5$; $\frac{S_2-z_2}{D} \leq 1.0$; $\frac{L}{D} \leq C56$ |
| 6 | $Q = \mu DW\sqrt{2g*(S_1-(z_2+D))}$ | $\frac{S_1-z_1}{D} \geq 1.5$; $\frac{S_2-z_2}{D} \leq 1.0$; $\frac{L}{D} > C56$ |

With: $Q$, the discharge through the culvert, $W$, the width of the culvert, $D$, the culvert height, $\mu$, the total head loss coefficient, $S_1$, water level on side 1, $S_2$, the water level on side 2, $h_1$, the water level above the culvert bottom on side 1, $h_2$, the water level above the culvert bottom at side 2, $h_c$, the critical water level inside the culvert, $z_1$, level of the bottom of the culvert at side 1, $z_2$, the level of the bottom of the culvert at side 2, $L$, the culvert length, $C56$, coefficient to differentiate between flow type 5 and 6.

### 2.3.3. Different Head Loss Coefficients

The in- and outflow culverts of the FCAs along the Scheldt estuary have additional features, like trash screens and non-return valves, etc., that have to be taken into account when calculating the discharge through these culverts. Most of these features can be accounted for by adding an extra head loss coefficient $C$ to the formulation of the discharge coefficient $\mu$ in Equation (1). The following head loss coefficients are important for modelling in- and outflow culverts for an FCA with and without CRT in the Scheldt estuary:

(1) The entrance head loss represents the head loss due to the contraction of the flow at the entrance of the culvert. An abrupt contraction at the culvert entrance causes a head loss due to the deceleration of the flow immediately after the vena contracta. A head loss coefficient $C_1$, of which the value is a function of the diameter ratio after and before the contraction, is proposed by [48]. For a culvert between a river and a floodplain, the contraction can be seen as very large, estimating the entrance head loss coefficient to be 0.5 according to [48]. Bodhaine [46] noticed that the entrance head loss coefficient $C_1$ for flow type 5 had to be lowered comparatively with the other flow types. The calculated discharge seemed to be overestimated when the default equation was used. Therefore, a correction coefficient $C5$ is multiplied with entrance head loss coefficient $C_1$ when flow type 5 occurs. An exact value for $C5$ is not given but according to Bodhaine [46] this coefficient lies in the following interval: $4 \leq C5 \leq 10$.

(2) The head loss due to pillars inside the culvert: Sometimes, at the entrance of culverts, the flow is divided into two sections by a pillar. This pillar causes additional head loss and is taken into account. According to [48], the head loss coefficient $C_p$ to account for a pillar is given by:

$$C_p = \beta\left(\frac{L_p}{b}\right)^{4/3}\sin\theta \tag{3}$$

where $L_p$ is the thickness of the pillar; $b$ is the distance between two consecutive pillars; and $\beta$ is a coefficient dependent on the cross-sectional area of the pillar and according to Bodhaine [46] $\beta$ equals 2.42 for rectangular pillars and 1.67 for rounded pillars; $\theta$ stands for the angle of the pillar with the horizontal plane. In most cases, this angle will be 90° and $\sin\theta$ will be equal to 1.

(3) The head loss due to internal friction: The head loss coefficient $C_2$ takes the head loss inside the culvert due to internal friction into account and is calculated according to [49]:

$$C_2 = \frac{2gLn^2}{R^{4/3}} \tag{4}$$

where $L$ is the length of the culvert; $n$ is the Manning Strickler roughness coefficient; $R$ is the wet cross-sectional area in the culvert.

(4)  The exit head loss: $C_3$ represents the head loss coefficient due to expansion of the flow exiting the culvert. It is calculated according to [49]:

$$C_3 = \left(1 - \frac{A_s}{A_{s2}}\right)^2 \tag{5}$$

where $A_s$ and $A_{s2}$ are the sections just in- and outside the downstream end of the culvert.

(5)  The head loss due to non-return or one-way valve: All outflow culverts have non-return valves on the estuary side to prevent water from entering the FCA (see number 1 in Figure 2). Depending on the opening, the valve will cause more or less head loss. $C_V$ represents the head loss coefficient due to the presence of a non-return valve. For a flap gate valve rotating around hinges at its upper edge, values for $C_V$ were obtained experimentally by [48]. Four values for $C_V$ are given in Table 3 according to the opening of the valve.

like for head loss coefficient $C_1$, a correction coefficient $C_V5$ is multiplied with the head loss coefficient $C_V$ to take into account the increase of the head loss when applying flow type 5. Through a number of laboratory experiments with a physical scale-model at Flanders Hydraulics Research, the value for this coefficient was determined to be 1.5 [45].

(6)  The head loss due to the presence of a trash screen: Trash screens in front of the inflow and outflow culverts prevent garbage, drift wood, and plant debris from clogging the culverts (indicated by number 4 in Figure 2). The head loss due to the presence of these screens can be estimated by its relationship with the velocity head through the net flow area. The head loss coefficient $C_T$ accounting for the presence of a trash screen can be calculated according to [50]:

$$C_T = \left(1.45 - 0.45 A_T - A_T{}^2\right) \tag{6}$$

where the ratio of net flow area, $A_{net}$, to gross rack area, $A_{gross}$, is given by $A_T$:

$$A_T = \frac{A_{net}}{A_{gross}}. \tag{7}$$

(7)  Wooden beams in front of the inflow culvert to function as a small weir: The height of these wooden beams is used to fine tune the moment the flow enters the FCA during flood in the estuary (indicated by number 2 in Figure 2). This structure will not be taken into account with the head loss. Instead, on the side where this wooden weir structure is present, the bottom level of the culvert will be set equal to the top of this wooden weir. For the entrance diameter or opening of the culvert, the height of the small weir will be subtracted from the height of the culvert. This structure makes the overall modelling of the culvert discharge more complicated. However, this assumption provides the correct time of water inflow in an FCA with CRT in the calculations.

(8)  Downward sliding valves to close the culvert: Sliding valves were designed to close the culvert for maintenance or to prevent inflow in an FCA with CRT in case of a storm surge. However, in practice, these valves are often used to smother the inflow of the culverts (indicated by number 3 in Figure 2). No additional head loss coefficient is defined for these valves. The length over which these valves are let down is subtracted from the culvert height in the calculations.

**Table 3.** Values for the head loss coefficient $C_V$ depending on the opening of a flap gate valve according to [48].

| Valve Position | Wide Open | $\frac{3}{4}$ Open | $\frac{1}{2}$ Open | $\frac{1}{4}$ Open |
|---|---|---|---|---|
| $C_V$ | 0.2 | 1. | 5.6 | 17 |

　　　Each in- and outflow construction at an FCA with or without CRT in the Scheldt estuary usually consists of multiple culverts placed next to each other. The geometric properties along with all the head loss coefficients have to be provided by the user for each culvert to the software. All culvert parameters depend on the physical properties of the specific culvert construction and are independent of water levels around the culverts. At each time step, a Fortran subroutine that contains the equations for the different flow types and the conditions when they occur, according to Table 2, named buse.f, is called. At each time step, the discharges are calculated for all the culverts given by the user. The calculated discharges are then further used as sink and source terms for the in- and outflow points, respectively, of the culverts. Appendix A offers more detailed information on the practical implementation of all the equations and conditions into the TELEMAC subroutine buse.f. This subroutine is used by the 2D and the 3D module. The difference is that in TELEMAC-3D, the water is actually added or taken from the water column (model layer) at the specific height corresponding with the culvert bottom level. All the geometric parameters and head loss coefficients for each culvert are listed in a text file that is read by a subroutine called lecbus.f. Examples are given in the following sections.

### 2.4. FCA with CRT Bergenmeersen: Detailed 3D Hydrodynamic Model

　　　With this detailed 3D hydrodynamic model of this FCA, we want to demonstrate that the implementation of the culvert flow into TELEMAC is able to reproduce the discharges through the culvert construction of this FCA. First, the FCA will be introduced in detail, followed by a detailed model, and chosen culvert parameters description. Field measurements of discharges through the culverts will be used to calibrate the culvert parameters. Finally, the validation will be done by simulation the storm surge of 6 December 2013, where modeled and measured water levels inside the FC will be compared.

### 2.4.1. FCA with CRT Bergenmeersen

　　　Bergenmeersen is the name of an FCA with CRT and is located upstream in the Scheldt estuary at 153 km from the estuary mouth (see Figure 1 for location in the estuary). The FCA covers 41 ha and became operational in the summer of 2013. The tide enters semi-diurnal through six inflow culverts. The inflow culverts are built on top of the outflow culverts as can be seen in the cross section of the construction in the sketch in Figure 2. Trash screens are present in front of the entrance to prevent debris from clogging the culverts. The six inflow culverts are all 2.7 m wide and 2.2 m high. Their bottom level lies at 4.2 m TAW (where 0 m TAW corresponds to low water level at sea) and their length is 9.5 m. There are six outflow culverts situated under the inflow culverts. On the FCA side, trash screens are also present to prevent clogging of the culverts by floating remnants of plants from the FCA. These outflow culverts are 1.35 m wide and 1.1 m high. Their bottom level lies at 2.7 m TAW and they are 18 m long. A wide creek was dug central in the FCA. The material from this excavation was used to heighten the ground level of the FCA near the flood protecting ring dike at the North side (see Figures 1b and 3), as this was the lowest part of the FCA. This was done to prevent stagnant water and the accompanying mosquito problems for the neighboring houses. There are two older outflow culverts on the East. These culverts are connected with a ditch to the central creek. These are also square culverts, 1.5 m high and wide. Their bottom level lies at 2.5 m TAW. A third older outflow culvert is located on the West side. The dimensions of this culvert correspond with the other two older outflow culverts, but the bottom level of this one lies at 3.0 m TAW. The locations of all these culverts and the central creek are shown in Figure 1b. Almost the entire dike between the Scheldt estuary and the flood plain is constructed as overflow dike. The crest level of this overflow dike varies between 6.3 and 6.4 m TAW [51]. The crest level of the flood protecting ring dike is 8 m TAW.

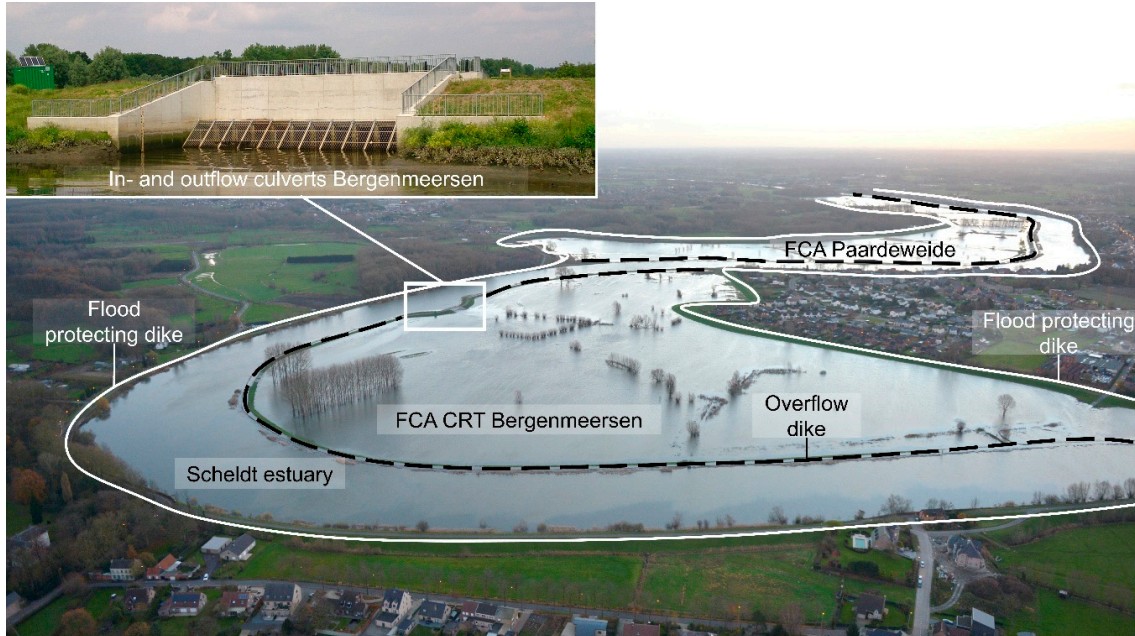

**Figure 3.** Helicopter photo of FCA Bergenmeersen on the morning of 6 December 2013, one hour after high water level of the 'Sinterklaas' storm surge (6/12/2019 09:25 local time). The thin white line indicates the location of the flood protecting dike and the thicker black dashed line indicates the location of the overflow dike. The panel in the top left corner gives a closer view on the in- and outflow culvert construction seen from within the FCA when the water level is low. The trash screens are visible.

2.4.2. Detailed 3D Hydrodynamic Model

Figure 1b shows the domain plan view of a detailed 3D hydrodynamic model together with the topography of this FCA. It was cut out of the bigger SCALDIS model and the mesh was refined to a resolution of 7 m. All other model parameters were kept from the SCALDIS model. The detailed Bergenmeersen model is driven by a water level boundary downstream and a discharge boundary upstream. The water levels used at the boundary were measured water levels just outside the in- and outflow culvert construction. The upstream discharge boundary is set to a fixed 1 m$^3$/s discharge. In September 2014, a 13-h measurement campaign was executed (13 h to capture one full tidal cycle), measuring the in- and outflow discharges with a StreamPro ADCP. Divers measured the water level in- and outside the FCA. These measurements are used to calibrate the input parameters for the culverts of FCA Bergenmeersen. A detailed description of all model parameters needed to model the discharge through culverts is given in Appendix A. An overview of the culvert parameter values after calibration is given in Appendix B Table A1. This 3D model of FCA Bergenmeersen is also available in the TELEMAC-MASCARET software package as a validation case.

In this section, the choices for the culvert parameter values are described. The culvert entrance head loss coefficient $C_1$ was chosen equal to 0.5 according to [48] when a large contraction is assumed going from the river to the culvert. The correction factor for the entrance head loss coefficient for flow type 5 was chosen equal to 6 after calibration and given the interval proposed by Bodhaine [46]. The inflow culverts have squared pillars in the middle and $C_p$ is calculated according to Equation (3) where $L_p$ equals 0.35 m; $b$ equals 1.35 m; and $\beta$ equals 2.42 for rectangular pillars. The angle between the pillar and the horizontal is 90° so that $\sin \theta$ equals 1. With these values, $C_p$ equals 0.4. In the TELEMAC code, there is no variable foreseen for $C_p$, so its value will be added to the entrance head loss coefficient for the entrance of the inflow culverts at the river side only. The head loss due to internal friction within the culvert is calculated by the model by giving the length of the culvert and a Manning Strickler friction coefficient to solve Equation (4). The length of the culvert construction is 18 m, but given the specific construction type, this value is brought back to 1 and 9 m for the in- and outflow culverts, respectively.

For the Manning Strickler coefficient, a value of 0.015 s/m$^{1/3}$ is taken corresponding to smooth concrete. For the exit head loss coefficient, Equation (5) suggests that for a very large expansion, the coefficient would be close to 1, but taking the neighboring outflow culverts into account, the same value as the entrance head loss was chosen: 0.5. The value 1 was kept for the inflow culverts. Non-return valves are present on the outflow culverts. These are relatively lightweight high density polyethylene (HDPE) valves rotating around hinges on the top. The angle of opening was measured in the field for this type of valve and the measurements showed a maximum opening angle of 70° [52]. This corresponds to a head loss coefficient $C_v$ equal to 1 according to Table 3 and [48]. For the trash screens, the ratio $A_T$ is equal to 0.8, which makes $C_T$ equal to 0.45 according to Equation (6). This value was rounded to 0.5. Trash screens can sometimes have a lot of debris on them, creating more head loss. Therefore, the head loss coefficient for trash screen is usually used for calibration of the total head loss. In this calibration case, a day before the measurements the screens were cleaned, and thus the value of 0.5 was kept here. To differentiate between flow type 5 and 6, a coefficient $C56$ is chosen equal to 10 according to [46]. The width of the inflow culverts is 2.7 m. The width of the outflow culverts is 1.35 m. The level at which flow enters the inflow culverts depends on the bottom level of the inflow culvert and the height of possible stacked wooden weir logs in front of it. The total inflow area is further restricted possibly by the height of how far the sliding valves were let down. For the three older outflow culverts, the same parameters were chosen, but their geometric values were changed accordingly: Their width and height is 1.5 m. Their length is 30 m and their bottom level lies at 2.5 m TAW for the two in the East and 3.0 m TAW for the one in the West (for locations see Figure 1b).

The measured discharges of the 13-hour measurement campaign were used to calibrate the chosen culvert parameters. First, values for all culvert parameters are chosen from the guidelines given in Section 2.3.3 and are based on the culverts' geometry and elevation. According to Equation (1), all head loss coefficients are summed, so for calibration it does not matter which one is adapted. We used the trash screen head loss coefficient. More important than the head loss coefficients are the right geometry values of the construction in situ, which can vary from the values taken from as-built plans in the office. The culvert flow equations are sensitive for the exact water levels before and after the culvert. The topography of the FCA is very important to obtain correct water levels inside the FCA, and thus correct simulated discharges. We noticed that a ditch that connects the main creek with the two older outflow culverts in the East of Bergenmeersen was crucial for a good drainage of this part of the FCA and it was not well represented in the model topography (it is shown in Figure 1b). After improving the model topography for this ditch, much better results were obtained. Figure 4 shows the final calibration result with the measured discharges for in- and outflow on 22 September 2014 as yellow dots. The measured water levels inside the FCA and in the Scheldt are also given with the bottom levels of the in- and outflow culverts to understand better when in- and outflow starts and stops. The dotted lines give the modeled discharges. The difference between the measured and modelled discharges is given by the calculated root-mean-square error (RMSE) values. For the inflow discharges, the RMSE was 1.06 m$^3$/s and for the outflow discharges the RMSE was 0.57 m$^3$/s.

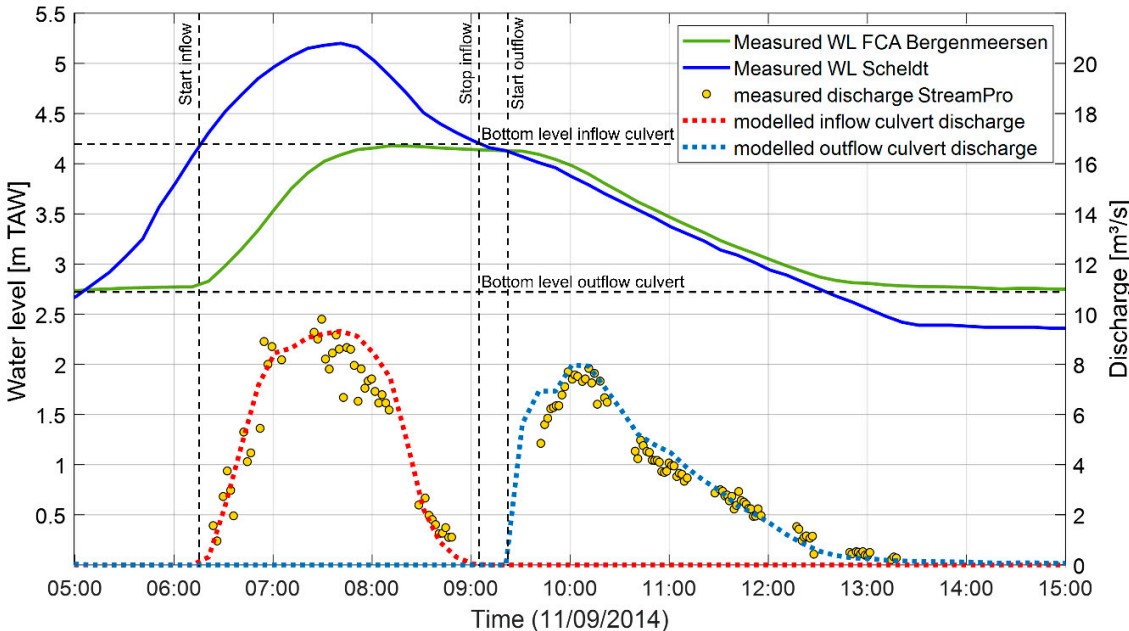

**Figure 4.** Calibrated modeled discharges compared to measured discharges through the in- and outflow culverts of FCA with CRT Bergenmeersen.

### 2.4.3. Validation of the Bergenmeersen Culverts Flow Model

To validate the culvert flow implementation in TELEMAC in this model, the storm surge of 6 December 2013 was used. The water levels measured just outside the in- and outflow construction on the Scheldt side, from 5 and 6 December 2013 were used as boundary conditions for the detailed 3D model of FCA Bergenmeersen. There are no discharge measurements from this flood event. Therefore, the calibrated culvert parameters from the previous section will be used without further changes. The validation will be done by comparing the measured and modeled water levels inside the FCA. If the discharges are simulated well, the water level inside the FCA should correspond well with the measured water level during the flood event.

### 2.5. Physical Scale Model

A scale model of four inflow culverts was built in a flume at Flanders Hydraulics Research to investigate the hydraulic jump inside this construction. However, the opportunity was taken to perform some extra measurements of discharges through the culverts of this model to use for extra validation of the culvert flow implementation in TELEMAC. A physical scale model has the advantage that it is a controlled environment, so water levels and discharges can be measured with greater precision compared to field measurements. First, the scale model is described, followed by the description of the different tests and the choices of the culvert parameters.

### 2.5.1. Scale Model Geometry

Four inflow culverts were built in a flume with a horizontal bottom. The water level upstream of the construction was controlled by the discharge of water added to the flume. The water level downstream was controlled by an adjustable weir. The height of this weir could be adapted to set the desired water level downstream of the culverts. The flume had a length of 34.80 m, a height of 0.75 m, and a width of 0.56 m.

A Froude scale factor of 15 was chosen in order to use the largest possible dimensions that fitted within the small flume. Four inflow culverts were implemented into the flume. Each culvert had a width of 0.087 m and a height of 0.147 m. The wall thickness between the culverts was 0.026 m. The total width was then 0.426 m, so on one side, the flume width was constraint to this total width.

Figure 5 gives a side view photo of the flume with the culverts and a schematic side and top overview of the flume. The discharges measured in the scale model were compared with calculated discharges with the new culvert code implemented in TELEMAC. Because water surface tension, air–water interaction, and friction could not be scaled down, the results were not scaled up to real life dimensions as Reynolds similarities could not be guaranteed. Therefore, it was chosen to compare only the scale model measurements with the modeled discharges through the scale geometry.

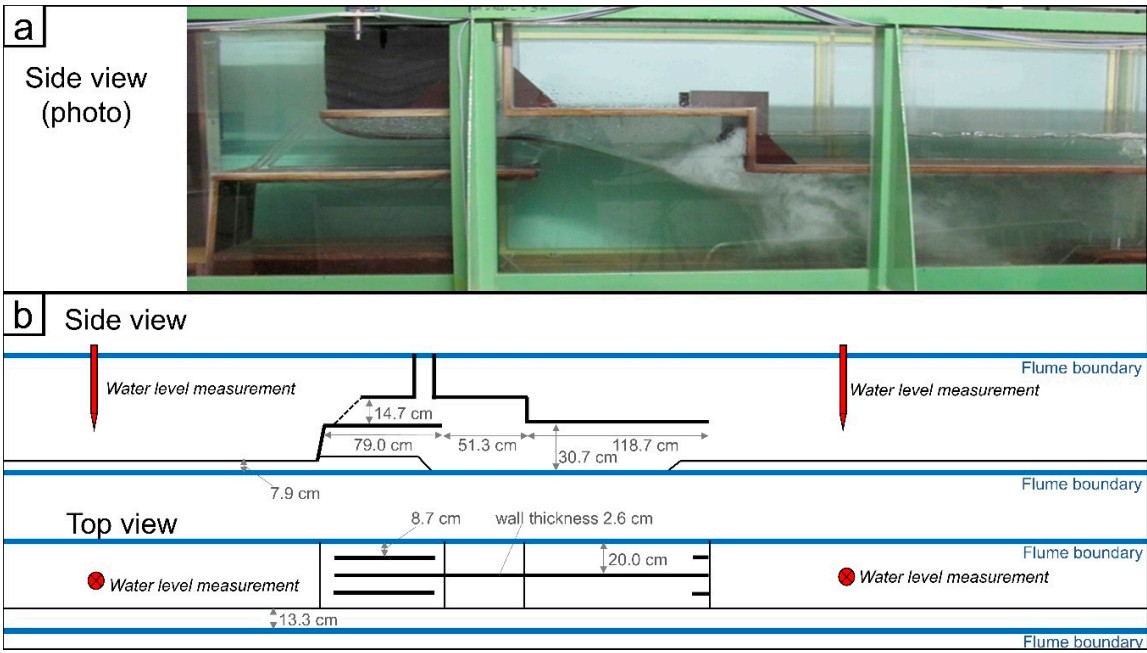

**Figure 5.** Side view photo of the actual scale model in the flume (**a**) and flume and scale model dimensions with side and top view plan (**b**). Water flows from left to right.

### 2.5.2. Scale Model Tests Setup

Water is added to the flume in an upstream reservoir. From this reservoir, it flows over a sharp crested weir into the main flume compartment. The sharp crested weir makes an accurate estimation of the discharge possible. A manual valve is used to stifle the flow and an electronic valve is used to tune the discharge. For the tests, specific water levels were chosen upstream and downstream of the culverts scale model. The discharge entering the flume is used to control the upstream water level. The downstream water level is controlled by a weir by changing the crest level. If both water levels (upstream and downstream of the culvert scale model) are set and a steady flow is reached, the water level measured in the reservoir and the dimensions of the sharp crested weir give an estimate of the discharge flowing through the culverts in the scale model. After 60 s of steady flow, a next test was started by setting new water levels up- and downstream the culverts scale model. Water levels in the flume were measured with a water level measuring needle. Three needles were used to measure water levels: One in the reservoir, and one up- and downstream of the scale model. Six sets of up- and downstream water levels were chosen, and the test was repeated for a scale model with a trash screen in front of the inflow culverts. The first five sets of water levels all have free outflow and the sixth has a submerged outflow. The water levels were chosen based on real life dimensions of the construction and were then scaled down. They are listed in Table 4.

**Table 4.** List of six sets of up- and downstream water levels for the scale model.

| Set # | Upstream Water Level | Downstream Water Level | Upstream Water Level | Downstream Water Level |
|---|---|---|---|---|
| | (Reality) | (Reality) | (Model) | (Model) |
| | [m TAW] | [m TAW] | [m] | [m] |
| 1 | 4 | 3 | 0.361 | 0.293 |
| 2 | 5 | 3 | 0.429 | 0.291 |
| 3 | 6 | 3 | 0.494 | 0.293 |
| 4 | 7 | 3 | 0.559 | 0.291 |
| 5 | 8 | 3 | 0.626 | 0.292 |
| 6 | 7 | 6 | 0.560 | 0.492 |

### 2.5.3. Culvert Parameters

An overview of all culvert parameters for these tests is given in Table A2 in Appendix B. The values chosen are explained here. The entrance head loss coefficient was estimated based on the vertical contraction and a value between 0.4 and 0.5 is found following [48]. The culverts were made out of plexiglass and a Manning Strickler friction coefficient of 0.012 $s/m^{1/3}$ was chosen. The exit head loss coefficient was set to 0.2 as there is not a big expansion downstream of the culvert in the flume. The width of the culvert was 0.087 m and the height was 0.147 m. The bottom level of the inflow culverts was constructed at 0.313 m from the flume bottom. The length of the inflow culverts was 0.6 m. To differentiate between flow type 5 and 6 the parameter $C56$ was equal to 10 according to [46]. $C5$ was chosen equal to 6 according to the value taken for the Bergenmeersen culverts as both structures are very similar.

For the second test, a trash screen was placed in front of the inflow culverts. The miniature trash screen is made from 1.25 mm thick stainless steel. The vertical bars in the grille are as wide as they are thick (0.00125 m). The trash screen was placed over the angled (45°) culvert walls at the entrance. The bars of the trash screen cover 14%–17% of the total opening of the inflow culvert depending on the water level (there was a broad horizontal bar present at half the culvert height necessary for the strength of the scale model trash screen). Using Equation (6), the head loss coefficient for the trash screen was set to 0.3 for the tests with the trash screen and set to zero for the tests without the trash screen. All other parameters were kept constant for both tests.

For the calculation of the discharges, only the equations were applied. No TELEMAC model was built. The equations were, however, used exactly as they are used in the source code of TELEMAC.

### 2.6. Hindcast of Storm Surge and Impact of FCA in the Scheldt Estuary

This whole paper is written about the implementation of culvert flow equations in TELEMAC to be able to model the water exchange between the Scheldt estuary and FCAs. In FCAs with CRT, every tide water enters the FCA through the culvert construction. However, these FCAs were and are being built to protect the estuary from storm surges. Therefore, the impact of FCAs on water levels in the estuary during storm tide is demonstrated with the SCALDIS model. The storm surge of the 'Sinterklaas' storm of 6 December 2013 is modeled because it was the first larger storm surge inside the estuary since the implementation of FCA with CRT Bergenmeersen and water levels inside the FCAs were measured during the storm tide. Boundary conditions for the downstream water level boundary were extracted from a continental shelf model. In the SCALDIS model, wind and atmospheric pressure effects were not explicitly taken into account, assuming that these effects play a minor role inside the estuary because of the limited fetch length and limited water depths in such a narrow and shallow estuarine setting. It was assumed that the storm surge was generated on the continental shelf (e.g., [53]) and that the surge was included in the water level boundary conditions (as in e.g., [54]). Measured daily averaged discharges were imposed on the upstream boundaries. For this storm, measured water levels are available along the entire estuary to which the modeled water levels will be compared.

Three scenarios will be simulated:

(1) Scenario 1 will hindcast the storm surge of 6 December 2013 as it was. All FCAs that were active at that time are active in the model. This scenario will tell how well the model can simulate this storm surge and it will be used as a reference to compare the other two scenarios with.

(2) Scenario 2 starts from scenario 1 for which the largest intertidal marsh area in the estuary (the so-called Drowned Land of Saeftinghe, for location see Figure 1) is removed from the model domain. Its effect on storm surge attenuation was already demonstrated in [19,55]. This scenario is added to compare the impact of a large natural marsh on storm surge attenuation within the estuary with the impact of several smaller FCAs. This marsh was removed from the model domain by increasing its bottom level to a point where it cannot be flooded anymore.

(3) Scenario 3 starts from scenario 2 for which all FCAs are removed from the model domain.

Water levels along the estuary for these three scenarios will be compared with each other and with the measured water levels.

## 3. Results

### 3.1. Bergenmeersen Detailed 3D Model Validation

Four tides were simulated of which the third one was the storm tide. The thick blue line in Figure 6 shows the measured water levels in the Scheldt just outside the in- and outflow culvert construction of FCA Bergenmeersen. The thin black line with the small black circles shows the water level measured inside the FCA near the in- and outflow culvert construction. The red line is the modeled water level inside FCA Bergenmeersen. The red line follows the measured water levels well, except for the end of the ebb tide after the storm surge, where the model slightly underestimates the water level. The difference between modeled and measured water levels was calculated with the RMSE and was equal to 0.08 m for the entire simulation, i.e., the four tides. The FCA drains faster in reality than in the model, although the two spring tides before the storm tide show a model that drains just a little faster than what was measured. The water level in the Scheldt estuary just reached the crest level of the overflow dike during the storm surge. The photo in Figure 3 also shows this. The water level is just at the level of the overflow dike, and at some places just overtopping the overflow dike. Inside the FCA, the water level remained half a meter below the crest level of the overflow dike. The measurements and the model show that the FCA filled with water that came through the inflow culverts.

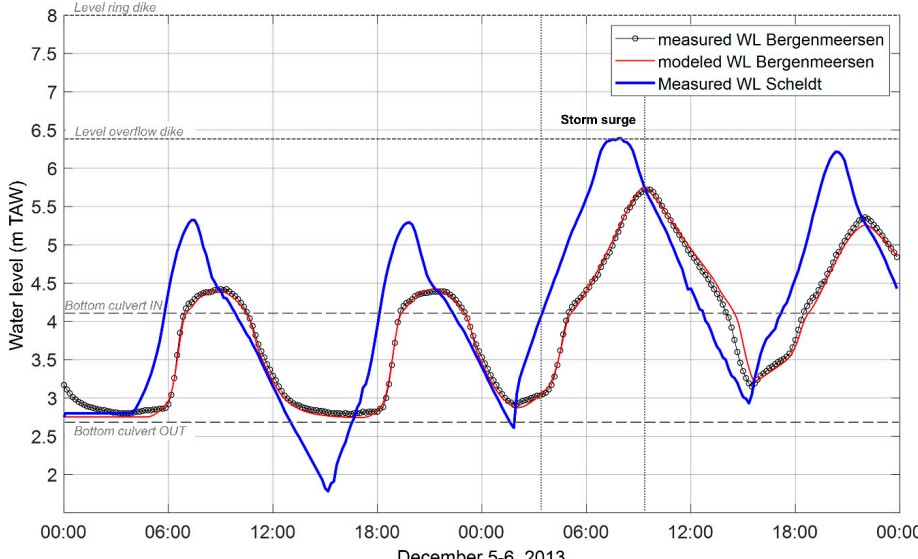

**Figure 6.** Comparing modeled and measured water level variations inside the FCA with CRT Bergenmeersen for the storm surge on 6 December 2013.

### 3.2. Physical Scale Model Tests

The measured discharges through the scale model in the flume and the corresponding calculated discharges are compared in Table 5. The results are numbered in the same order as shown in Table 4 and for clarity, the water levels on both sides of the culverts in the scale model are repeated. To compare both measured and calculated discharges, their difference is taken and expressed in percentages. In the first five sets, the water level downstream remains almost constant while the upstream water level rises. For these sets, the difference in discharges between measured and calculated is small except for the set number five with the highest water levels upstream. There, the calculated discharge is underestimated with almost 7% from the measured one. Set number six has a submerged outflow and this results in flow type 4. Here, the difference between modeled and calculated discharge is 1.6%.

**Table 5.** Results of flume scale model compared with the calculated discharges.

| Set # | Upstream Water Level | Downstream Water Level | No Trash Screen | | | Trash Screen | | |
|---|---|---|---|---|---|---|---|---|
| | Model [m] | Model [m] | Measured Q [m³/s] | Calculated Q [m³/s] | Difference [%] | Measured Q [m³/s] | Calculated Q [m³/s] | Difference [%] |
| 1 | 0.361 | 0.293 | 0.006 | 0.006 | 0 | 0.005 | 0.005 | 0 |
| 2 | 0.429 | 0.291 | 0.025 | 0.025 | 0 | 0.023 | 0.022 | −4.3 |
| 3 | 0.494 | 0.293 | 0.050 | 0.049 | −2 | 0.046 | 0.043 | −6.5 |
| 4 | 0.559 | 0.291 | 0.063 | 0.063 | 0 | 0.060 | 0.060 | 0 |
| 5 | 0.626 | 0.292 | 0.076 | 0.071 | −6.6 | 0.073 | 0.068 | −6.8 |
| 6 | 0.56 | 0.492 | 0.061 | 0.062 | 1.6 | 0.057 | 0.054 | −5.2 |

The discharges in the test with the trash screen are all lower than the discharges in the test without the trash screen because of the extra head loss. The calculated discharges have decreased more than the discharges measured through the scale model. Set number four and five have water levels resulting in flow type 5 and for these sets the difference remains almost the same. The first three sets have water levels resulting in flow type 2. At this flow type, the underestimation of the calculated discharges increased. For set number six with flow type 4, the difference evolved from an overestimation by the calculated discharges to a larger underestimation.

### 3.3. SCALDIS Estuary Scale Storm Surge Simulations

The results of the three scenarios and the measured high water levels of the storm surge are shown in Figure 7. It is important to note that the maximum water levels, shown in Figure 7 do not appear at the same moment in the estuary and so the slope of the lines in this figure does not represent a water level gradient. The thick blue line shows the maximum water levels generated by the model simulation of scenario 1, i.e., the hindcast of the storm surge level along the length axis of the estuary. The red diamonds indicate the measured maximum water levels during the storm surge. From km 25 to km 70, the modeled high water levels are lower than the measured ones, but still within an acceptable 10 cm difference. At the most upstream part, the water level is influenced largely by the upstream discharges. The use of daily averaged discharges might affect the water levels here. In the largest part of the estuary, the model succeeds well in reproducing the high water levels of the storm surge. For additional reference of the maximum water levels attained by the storm surge, the crest levels of the flood protecting dikes along the estuary are given on top of Figure 7. At the most upstream part of the estuary, the storm surge high water level remained about 0.7 m below the crest level of the flood protecting dikes.

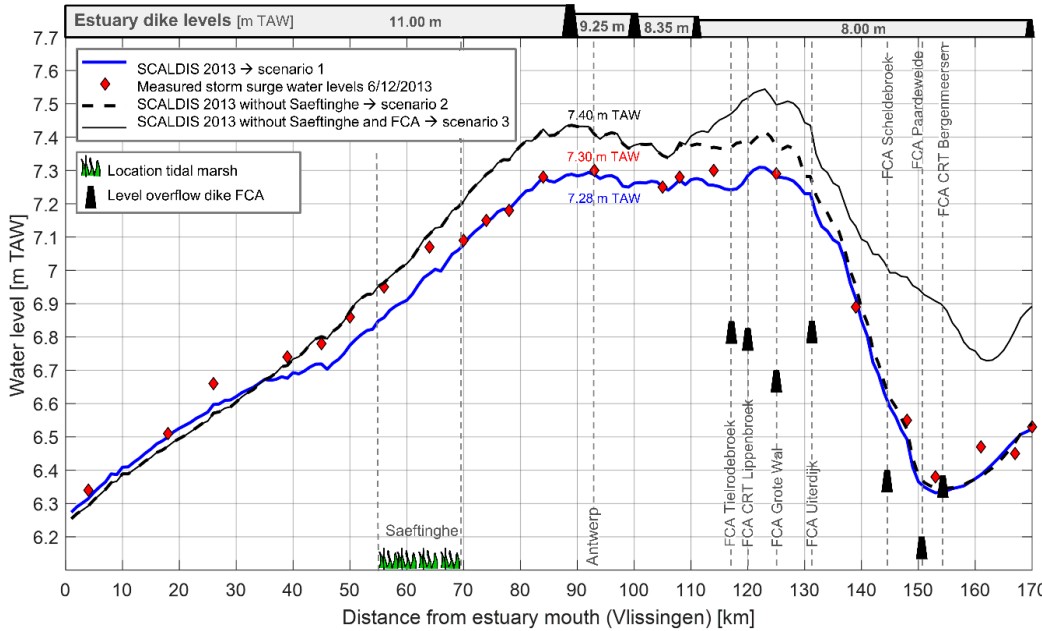

**Figure 7.** Water levels scenario simulations SCALDIS 2013 with indication of flood protecting dike and overflow dike levels.

In scenario 2, the large marsh of Saeftinghe was removed from the model. The location of this marsh is indicated in Figure 7. The location of all the active FCAs is also given by a trapezium symbol that indicates the level of their overflow dike. The thick black dashed line gives the high water levels for the storm surge without the large marsh area of Saeftinghe in the model. Starting at km 35 from the estuary mouth, the high water levels increase compared to the reference scenario; that is, 20 km downstream of the Saeftinghe marsh. The high water levels reach a maximum around Antwerp (km 90). Here, the simulated surge level is 10 cm higher when the large marsh area of Saeftinghe is excluded from the model domain (scenario 2) as compared to the simulation including the large marsh (scenario 1). However, in the upstream part of the estuary, the FCAs reduced the high water levels again to the level of the reference scenario 1.

In scenario 3, additionally to scenario 2, the FCAs were removed from the estuary. The thin black line shows the high water levels for this scenario. Up until km 110, they follow the high water levels of scenario 2. The first active FCA (Tielrodebroek) lies at km 118 from the estuary mouth. The effect of the FCAs on the high water level is large. The maximum difference can be seen around km 150 and is more than 0.5 m. In this scenario, around km 123, the storm surge water level reaches its maximum and only remains 0.45 m below the crest level of the flood protecting dikes in this area.

## 4. Discussion

### 4.1. Storm Surge Height Reduction by FCAs

FCAs and FCAs with CRT are very specific for the flood defense system of the Scheldt estuary. This hybrid system combines a nature-based function, i.e., the creation of natural marshes through reduced tidal inundation, with the engineered function of flood control. The CRT principle to restore natural marshes inside embanked areas is found in other places, like the UK [56], under the name of self-regulating tide (SRT), but is there not combined with a flood protection function. In terms of storm surge attenuation, a lot of work has been done to understand how a storm surge is attenuated over a coastal marsh (e.g., [14,15,17,18,54]) or by mangrove forests (e.g., [16,57]). Although this type of storm surge attenuation is more important for coastal regions, the FCAs in the Scheldt estuary function more as storage volumes. The specific height of the overflow dike for each FCA determines the timing when

storm surge water can enter as to efficiently use the available storage capacity. In contrast with natural marshes [19,31,39,58,59] or managed realignments [60,61], FCAs only use their storage capacity to extract the top of the tidal storm surge. The height of the overflow dike saves the storage capacity for when it is the most needed [31], making these areas more efficient in storing storm surge water.

For the hindcast of the 'Sinterklaas' storm (scenario 1) the SCALDIS model performed well, except for the region up- and downstream the large tidal marsh of Saeftinghe. An explanation could be that this model was not calibrated for storm surges and a large tidal marsh has little influence on daily tides when calibrating model. However, the large storage capacity during storm tides [18,19,55] could influence the tidal wave traveling along the estuary. The differences between measured and modeled surge levels remain well below 0.1 m and this is considered a good resemblance. The SCALDIS model uses daily averaged discharge values for its upstream boundaries. The averaging of these discharges causes a small mismatch between the modeled and measured high water levels upstream the estuary.

The importance of the large marsh of Saeftinghe for storm surge attenuation was already demonstrated before [19,55]. Removing it from the estuary (scenario 2) shows again the impact of this area on storm surge attenuation along the estuary. Around km 125, the water levels in scenario 2 start to move towards the water levels modeled in scenario 1. This shows that the FCAs in that part of the estuary are handling the extra storm water. The current modeled storm had a return period of 19 years [27], while the hybrid flood defense system was designed to withstand a storm with a return period of 4000 years [30], and although not all designated areas were built or active yet in 2013, it shows the ones that were active were far from their full capacity with this storm. The influence of the FCAs in storm surge attenuation is then shown in scenario 3 when they were also excluded from the simulation. The results show that different smaller FCAs along an estuary with optimal design for storage capacity are very effective in storm surge attenuation. In confined estuaries with little accommodation space to realign large areas for natural marsh development and flood mitigation, these smaller FCAs (with CRT) are a great solution as a hybrid flood defense system.

For storm surge modeling in the Scheldt estuary specific, these results demonstrate the importance of the implementation of the culvert flow equations into TELEMAC. The flood defense system, Sigmaplan [30], foresees more FCAs to be constructed in the future. One of the largest (600 ha) FCAs, partially with CRT, opened in the summer of 2017 and will further significantly reduce the flood risk for upstream locations along the estuary [26]. The SCALDIS model has proven its proficiency and is ready for scenario analysis with more severe storm surges and the impact of sea level rise and climate change.

### 4.2. Culvert Flow Implementation in TELEMAC

The implementation of the equations to calculate discharges through culverts for different flow types was done using a number of head loss coefficients that have a physical basis. This has the advantage that for culverts, where no discharge measurements are available, a good estimation of the head loss coefficients can still be made and thus a good estimation of the discharge through the culverts can be achieved. The implementation of the code in TELEMAC has a few limitations. One of these limitations is that the discharge is calculated per time step. Within this time step, water is extracted from the inflow side of the culvert in the model and is added at the outflow side of the culvert in the model. For longer culverts and smaller time steps, this immediate exchange of water is an approximation. The culverts in the FCAs in the Scheldt estuary are short (e.g., around 10 m) and with a time step of 5 s in the SCALDIS model, this posed no problem. Another limitation of the code is the fact that for larger culverts, the calculated discharge is sometimes larger than the amount of water that is available in the node of the model at that specific time step, especially in the case of a mesh with very fine resolution in combination with larger culverts. This limitation is hard coded into the subroutine handling the culverts and needs to be checked when calibrating or just modeling culverts. It could lead to a serious underestimation of the discharges in the model. A solution used in the SCALDIS model was to locally deepen the bathymetry of the nodes that represented the culverts.

A new option is also available to calculate the availability of water in the model on a node taking its surrounding nodes into account. This could also resolve this issue.

### 4.3. Detailed 3D Model FCA Bergenmeersen

After an initial calibration of the culvert parameters, the modeled water levels show very good agreement (RMSE of 0.08 m) with the measured water levels in FCA Bergenmeersen for the modeled storm surge. The results for the storm tide show that almost no water entered the FCA over the overflow dike. Before the storm, the inflow culverts were not closed with the sliding valves. From the helicopter photo (Figure 3) it seems that the FCA is almost completely filled with water and one can easily assume that it was because of storm water entering over the overflow dike. This small 3D model shows clearly that almost all water in the FCA entered through the inflow culverts. For larger storm surges, the sliding valves of the inflow culverts should be closed beforehand to maintain a larger storage capacity inside the FCA for stormwater entering over the overflow dike, and thus increase its efficiency.

In [39], the equations for the different flow types are used with fixed discharge coefficients derived from tests with a physical scale model. The implementation of the culvert equations into TELEMAC has strongly improved the way discharges through culverts of an FCA are calculated. In [39] a physical scale model was necessary to provide the right discharge coefficients, while now it is possible to give physically based head loss coefficients adapted for any shape and size of the culvert construction. Further, in [39], the water level inside the FCA had to be calculated based on the amount of water that already entered the area and the topography. Now, the free surface flow of the water in the FCA can be calculated by the TELEMAC software in 2D or 3D. For very small FCAs with only a single in- and outflow, a calculation by hand would still be feasible, but a detailed model of the FCA provides more insight in the complex filling and emptying. For the FCA Bergenmeersen, the detailed model showed how all of the storm surge water entered through the inflow culverts. It also showed that around 10% of the water entering the FCA daily exits through two older outflow culverts on the east side of FCA Bergenmeersen. This kind of information is important for optimal management of the FCA. Furthermore, the management of FCAs with CRT, where tidal habitat is restored [31,33,34,39], demands knowledge about nutrient fluxes, sediment balances, and estimations of flow velocities inside the culverts [62], and for this, the discharge data through the culverts are crucial. With the implementation of the culvert flow equations in TELEMAC, models can replace expensive and maintenance intensive ADCPs for the estimation of the discharges.

### 4.4. Physical Scale Model

Given the complexity of the culvert structure in the physical scale model, the differences between measured and calculated discharges were small (maximum difference was 6.8%). The calculated discharges for the test with the trash screen were underestimating the measured values and this might be due to the overestimation of the head loss coefficient for the scale model trash screen. This coefficient was calculated based on the physical properties of the trash screen (Equations (6) and (7)) in the scale model and was used as such in the calculation of the discharges. The physical scale model tests added a second and accurate dataset to which the new culvert code implemented in TELEMAC was validated.

Because Froude scaling was used for the scale model, but the Reynolds similarities could not be guaranteed for all flow conditions, the scale model results were not scaled up. Instead, the scale model results were compared to the results of the culvert flow equations applied with the scale model dimensions. It was not necessary to scale up the results to show that the culvert flow equations implemented in TELEMAC were able to reproduce the scale model results very well. The accurate measurements from the scale model must be seen as a second and independent dataset that validated the culvert flow implementation in TELEMAC.

## 5. Conclusions

This paper demonstrated the effectiveness in storm surge attenuation of a nature-based and engineered flood defense system in the Scheldt estuary. The combination of several FCAs along the estuary achieved a storm surge height reduction up to half a meter for the storm surge of 6 December of 2013. These results show the importance of these FCAs in the estuary and the necessity to incorporate them into an estuary scale 3D model for the assessment of storm surge reduction inside the estuary. Some of the FCAs are restored into tidal marsh ecosystems, by use of culvert constructions that allow daily reduced tidal in- and outflow. The implementation of model formulations for detailed calculation of the flow through culverts of these FCAs was an important aspect of this work. The comparison of simulated and measured discharges through a physical scale model of a culvert, and through a real scale culvert of an existing restored marsh during the storm surge, proves that the culvert flow equations with physically based head loss coefficients work well.

**Author Contributions:** Conceptualization, S.S., M.J.T. and A.L.; formal analysis, S.S., M.J.T. and T.M.; investigation, S.S.; methodology, S.S., A.L., P.M. and S.T.; resources, S.S. and A.L.; software, S.S., M.J.T. and A.L.; supervision, P.M. and S.T.; validation, S.S. and T.M.; writing—original draft, S.S.; writing—review and editing, M.J.T., T.M., P.M. and S.T. All authors have read and agreed to the published version of the manuscript.

**Funding:** This research received no external funding.

**Acknowledgments:** This research would not have been possible without the contribution of the people at Flanders Hydraulics Research who perform the measurements in the field, and especially those who contributed to the 13-hour measurement campaigns. A special thank u to them! The authors would also like to thank Tim Spiesschaert and Stefan Geerts for their help with the physical scale model and Jeroen Vercruysse for his advice.

**Conflicts of Interest:** The authors declare no conflict of interest.

## Appendix A. Practical Implementation of the Culvert Equations into TELEMAC

The TELEMAC source code is written in Fortran and already used a subroutine for the calculation of discharge through culverts. Only two types of flow were implemented: Free surface flow and pressurized flow (flow types 3 and 6). The subroutine is called buse.f and can be found between the subroutines of TELEMAC-2D. It is also called TELEMAC-3D. It was logical to use this subroutine and expand it with the five types of flow summarized in Table 2. To ensure backward compatibility and still expanding the same subroutine with more types of flow, a variable was created to let the user choose between the new and old way to calculate the discharge through culverts. This variable is called *OPTBUSE*. TELEMAC uses a list of key words in a steering file to read user choices for parameters values into the simulation. The keyword to give the variable *OPTBUSE* its value is *OPTION FOR CULVERTS* (value equal to 1 for the existing set of equations and value equal to 2 for the new set of equations distinguishing between five flow types).

The existing capability of TELEMAC to set source and sink terms anywhere in the domain was useful to implement culverts. The inflow and outflow of a culvert then act as a pair of sink and source points. For instance, when the flow is going from the river to the floodplain side, a source term is added on the floodplain side (positive discharge), and at the same time a sink term is set in the river (negative discharge). This implies the assumption that water leaving the river, enters the floodplain in the same time step. The length of the culvert and the time of a water particle to travel through the culvert are not taken into account. The discharges that are computed in the subroutine buse.f are simply added at the end of the existing sources matrix as follows:

$$QSCE2(NPTSCE + I) = -DBUS\%R(I) \tag{A1}$$

$$QSCE2(NPTSCE + NBUSE+I) = DBUS\%R(I) \tag{A2}$$

where *QSCE2* is the matrix containing the discharges through the point sources; *NPTSCE* is the number of point sources; I counts the number of culverts; *NBUSE* is the number of culverts; and *DBUS(I)* is the discharge computed in subroutine buse.f for culvert number I. This is done in the main subroutine for

TELEMAC-2D (called telemac2d.f). The same is done in the main subroutine for TELEMAC-3D (called telemac3d.f). The total number of source points, given by the variable *NSCE*, is now the sum of the number of sources (*NPTSCE*) and twice the number of culverts (a sink and source per culvert).

The equations and the conditions when to use which equation, summarized in Table 2 were implemented in the subroutine buse.f. The flow chart in Figure A1 shows how the code is built up once the *OPTION FOR CULVERTS* (*OPTBUSE*) is chosen.

The user has to specify all the parameters related to the culverts' dimensions and head loss coefficients in a separate text file. This file is recognized by the program through the keyword *CULVERTS DATA FILE*. This text file will be read by the subroutine lecbus.f. Both the text file and the subroutine lecbus.f already existed but were extended to take extra variables into account. The first and third line of the text file are comment lines and these are not read. On the second line, the first variable is the relaxation parameter (*RELAXB*). This relaxation parameter will give a weight to the discharge calculated at the current time step. This is a value between 0 and 1. The result is a weighted averaged discharge based on the discharge of this and the previous time step. After the relaxation parameter, there is a number indicating the number of culverts. The number of culverts needs to be given in the steering file through the keyword *NUMBER OF CULVERTS* (*NBUSE*) and this number will be checked with the number in the text file. The third line is commented and contains the names of all the parameters used in the subroutine buse.f. They are separated by a tab. The following parameters must be listed in the culverts data file:

- *I1*: mesh node number of culvert on side 1;
- *I2*: mesh node number of the culvert on side 2;
- *CE1*: entrance head loss coefficient for the culvert on side 1 (this corresponds to the head loss coefficient $C_1$);
- *CE2*: entrance head loss coefficient for the culvert on side 2 (this corresponds to the head loss coefficient $C_1$);
- *CS1*: exit head loss coefficient for the culvert on side 1 (this corresponds to the head loss coefficient $C_3$);
- *CS2*: exit head loss coefficient for the culvert on side 2 (this corresponds to the head loss coefficient $C_3$);
- *LARG*: the width of the culvert;
- *HAUT1*: height of the culvert on side 1;
- *CLP*: coefficient to restrict the flow direction (0 both directions are possible; 1 = only flow from side 1 to 2; 2 = only flow from side 2 to 1; 3 = no flow);
- *L*: linear head loss coefficient used only when *OPTBUSE* = 1; If *OPTBUSE* = 2, *L* is calculated;
- *RD1*: culvert bottom elevation on side 1 ($z_1$);
- *RD2*: culvert bottom elevation on side 2 ($z_2$);
- *CV*: head loss coefficient when a valve is present;
- *C56*: factor to differentiate between flow types 5 and 6;
- *CV5*: correction factor for $C_V$ when flow type 5 is used;
- *C5*: correction factor for $CE_1$ and $CE_2$ with flow type 5;
- *TRASH*: head loss coefficient when trash screens are present;
- *HAUT2*: height of the culvert on side 2;
- *FRIC* Manning Strickler friction coefficient;
- *LONG*: length of the culvert;
- *CIR*: indicates whether the culvert is rectangular (=0) or circular (=1); in case of a circular culvert the height is taken to calculate the wet section.

The computed discharge (*DBUS*) is positive for the flow from side 1 to side 2. The discharge through the culvert is computed in the subroutine buse.f according to the equations and conditions

given in Table 2 and given the parameters for the culverts given in the *CULVERT DATA FILE*. Two extra assumptions were made when calculating the discharges. The first one is that the critical water level ($h_c$) inside the culvert is assumed to be equal to two thirds of the culvert height. Secondly, when a wooden weir log is present in front of a culvert, an equivalent culvert bottom elevation is used replacing both the bottom elevations $z_1$ and $z_2$ in the equations itself (not in the conditions to determine when which type of flow is occurring). The mean between $z_1$ and $z_2$ is taken as equivalent bottom elevation of the culvert. The diameter of the culvert used in the equations will be the one corresponding to the side were the flow enters the culvert. For the wet section of the culverts, the shape of the culverts is taken into account. Rectangular and circular shapes are both possible. Other shapes are not foreseen in the code.

After calculation of the discharges through the culvert, first the relaxation of the calculated discharge of the current time step is computed. Based on the weight and the difference with the discharge in the previous time step, this can change the computed discharge of the current time step. Then, a test is performed to check if there is enough water present in the computational node at the current time step to extract the discharge from. A maximum of 90% of the available water is allowed to leave. Finally, the code checks that the culvert configuration allows the water to flow in the computed direction (see *CLP* variable), and if not, blocks the flow by setting the discharge to zero. This can be the case when a culvert has non-return or one-way valve at one side and this valve prevents the flow through the culvert.

The flow velocities exiting the culvert are calculated using the calculated discharge divided by the wet section. The direction of the flow velocity follows the direction of the culvert axis.

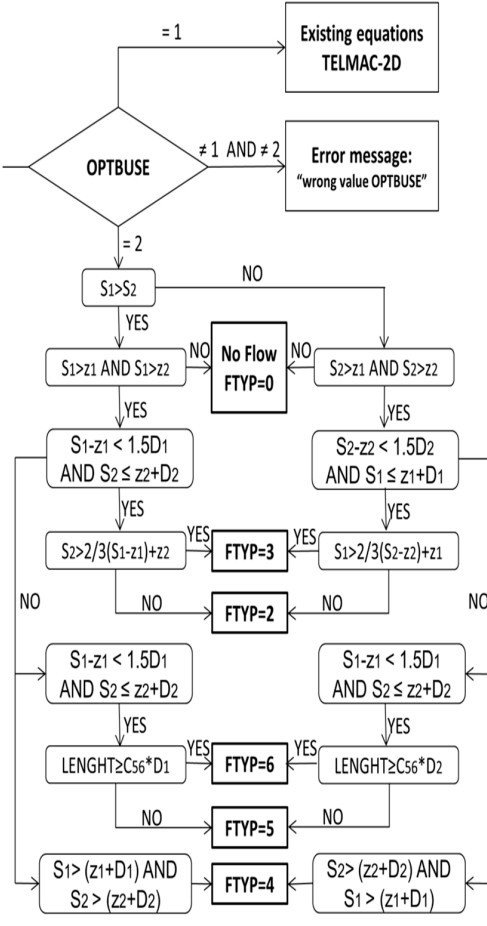

**Figure A1.** Flow chart showing the conditions for every type of flow in the buse.f subroutine.

It is possible to use passive or active tracers (like salinity, temperature, or sediments) when using culverts. The following equation describing the evolution of tracer concentration (*T*) is solved:

$$\frac{\partial T}{\partial t} + U\frac{\partial T}{\partial x} + V\frac{\partial T}{\partial y} + W\frac{\partial T}{\partial z} = v_t \Delta(T) + Q' \tag{A3}$$

where *t* is the time; *U*, *V*, and *W* are the flow velocity in the *x*, *y*, *z* direction, respectively; $v_t$ is the tracer diffusion coefficient; and $Q'$ is the source terms for tracers. In order to take the transport of tracers into account when using culverts, the user only has to specify the keywords related to the tracers in the steering file.

The subroutines buse.f and lecbus.f are both called from TELEMAC-2D and TELEMAC-3D in the same way. The difference between both is that in 3D the water and the tracer are extracted from the layer closest to the bottom level of the culvert.

Finally, Figure A2 shows an example of a modeled spring tide with the detailed Bergenmeersen model. This example shows for the different stages of the tide, i.e., the water level in- and outside the FCA, the different flow types that were used to calculate the discharges. Note that for the outflow culverts, the conditions to determine the type of flow do not take the non-return valve into account and so flow types are shown even if no flow is going through the culvert because of this valve. Negative flow types are given for outflow and positive for inflow. Note also that the inflow culverts experience some outflow for a short period of time after high water. This occurs when the water levels inside the FCA with CRT rise above the bottom level of the inflow culverts. The inflow culverts do not have non-return valves to restrict the flow from inside to outside the FCA.

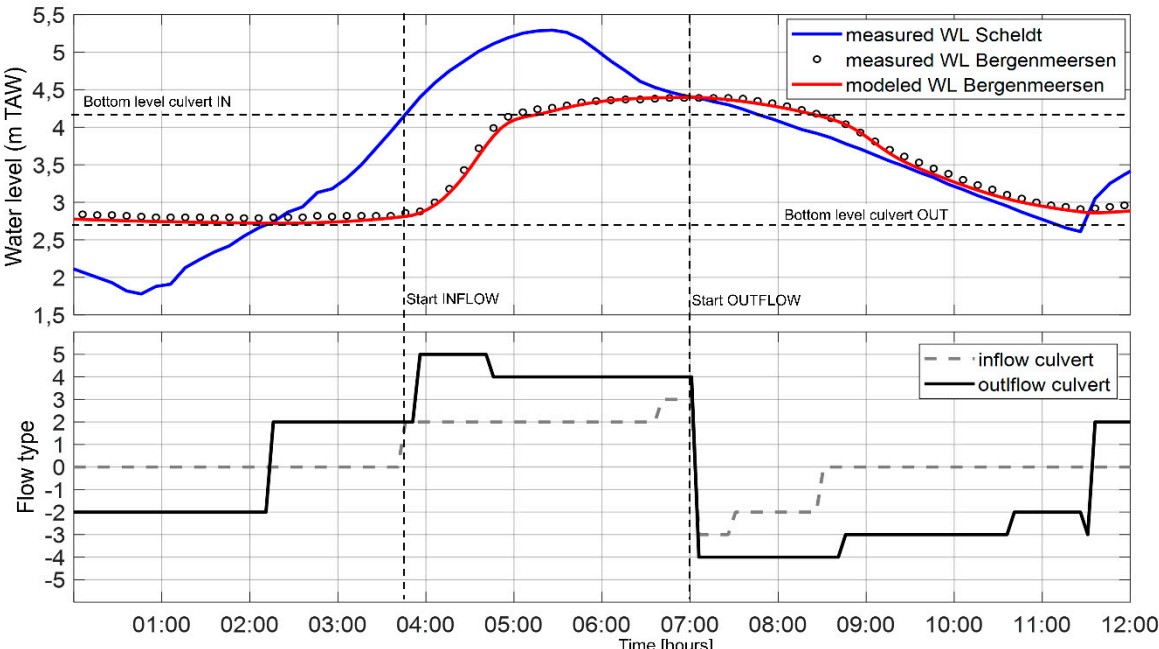

**Figure A2.** Different flow types for flow through culverts during spring tide at FCA with CRT Bergenmeersen.

## Appendix B. Model Parameters Culverts

This appendix gives an overview of the culvert parameters that were used in the different models or applications in this paper. The calibrated culvert parameters used in the detailed FCA with CRT Bergenmeersen model are given in Table A1. The culvert parameters that were used to calculate the discharges through the scale model are given in Table A2. These are the values for the scale model without a trash screen in front of the inflow culverts. For the calculations of the discharges through the

scale model with a trash screen, only the value of the head loss coefficient for trash screens (*TRASH*) changes from 0 to 0.3. All other parameter values remain the same.

**Table A1.** FCA with CRT Bergenmeersen culvert parameter values for use in buse.f.

| Parameter | Inflow Culverts | | | | | | Outflow Culverts | | | | | |
|---|---|---|---|---|---|---|---|---|---|---|---|---|
| | 1 | 2 | 3 | 4 | 5 | 6 | 1 | 2 | 3 | 4 | 5 | 6 |
| *CE1* | 0.9 | 0.9 | 0.9 | 0.9 | 0.9 | 0.9 | 0.5 | 0.5 | 0.5 | 0.5 | 0.5 | 0.5 |
| *CE2* | 0.5 | 0.5 | 0.5 | 0.5 | 0.5 | 0.5 | 0.5 | 0.5 | 0.5 | 0.5 | 0.5 | 0.5 |
| *CS1* | 1 | 1 | 1 | 1 | 1 | 1 | 0.5 | 0.5 | 0.5 | 0.5 | 0.5 | 0.5 |
| *CS2* | 1 | 1 | 1 | 1 | 1 | 1 | 0.5 | 0.5 | 0.5 | 0.5 | 0.5 | 0.5 |
| *LARG* | 2.7 | 2.7 | 2.7 | 2.7 | 2.7 | 2.7 | 1.35 | 1.35 | 1.35 | 1.35 | 1.35 | 1.35 |
| *HAUT1* | 0.35 | 0.35 | 0.35 | 0.45 | 0.25 | 0.35 | 1.1 | 1.1 | 1.1 | 1.1 | 1.1 | 1.1 |
| *CLP* | 0 | 0 | 0 | 0 | 0 | 0 | 2 | 2 | 2 | 2 | 2 | 2 |
| *RD1* | 4.5 | 4.5 | 4.35 | 4.2 | 4.2 | 4.2 | 2.7 | 2.7 | 2.7 | 2.7 | 2.7 | 2.7 |
| *RD2* | 4.2 | 4.2 | 4.2 | 4.2 | 4.2 | 4.2 | 2.7 | 2.7 | 2.7 | 2.7 | 2.7 | 2.7 |
| *CV* | 0 | 0 | 0 | 0 | 0 | 0 | 1 | 1 | 1 | 1 | 1 | 1 |
| *C56* | 10 | 10 | 10 | 10 | 10 | 10 | 10 | 10 | 10 | 10 | 10 | 10 |
| *CV5* | 0 | 0 | 0 | 0 | 0 | 0 | 1.5 | 1.5 | 1.5 | 1.5 | 1.5 | 1.5 |
| *C5* | 6 | 6 | 6 | 6 | 6 | 6 | 6 | 6 | 6 | 6 | 6 | 6 |
| *TRASH* | 0.5 | 0.5 | 0.5 | 0.5 | 0.5 | 0.5 | 0.5 | 0.5 | 0.5 | 0.5 | 0.5 | 0.5 |
| *HAUT2* | 1.6 | 1.6 | 1.6 | 1.6 | 1.6 | 1.6 | 1.1 | 1.1 | 1.1 | 1.1 | 1.1 | 1.1 |
| *FRIC* | 0.015 | 0.015 | 0.015 | 0.015 | 0.015 | 0.015 | 0.015 | 0.015 | 0.015 | 0.015 | 0.015 | 0.015 |
| *LONG* | 1 | 1 | 1 | 1 | 1 | 1 | 9 | 9 | 9 | 9 | 9 | 9 |
| *CIR* | 0 | 0 | 0 | 0 | 0 | 0 | 0 | 0 | 0 | 0 | 0 | 0 |

**Table A2.** Culvert parameter values for the scale model without a trash screen.

| Parameter | Inflow Culverts | | | |
|---|---|---|---|---|
| | 1 | 2 | 3 | 4 |
| *CE1* | 0.5 | 0.5 | 0.5 | 0.5 |
| *CE2* | 0.5 | 0.5 | 0.5 | 0.5 |
| *CS1* | 0.2 | 0.2 | 0.2 | 0.2 |
| *CS2* | 0.2 | 0.2 | 0.2 | 0.2 |
| *LARG* | 0.087 | 0.087 | 0.087 | 0.087 |
| *HAUT1* | 0.147 | 0.147 | 0.147 | 0.147 |
| *CLP* | 1 | 1 | 1 | 1 |
| *RD1* | 0.313 | 0.313 | 0.313 | 0.313 |
| *RD2* | 0.313 | 0.313 | 0.313 | 0.313 |
| *CV* | 0 | 0 | 0 | 0 |
| *C56* | 10 | 10 | 10 | 10 |
| *CV5* | 0 | 0 | 0 | 0 |
| *C5* | 6 | 6 | 6 | 6 |
| *TRASH* | 0 | 0 | 0 | 0 |
| *HAUT2* | 0.147 | 0.147 | 0.147 | 0.147 |
| *FRIC* | 0.012 | 0.012 | 0.012 | 0.012 |
| *LONG* | 0.6 | 0.6 | 0.6 | 0.6 |
| *CIR* | 0 | 0 | 0 | 0 |

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
