# Peer review of "Modeling Storm Surge Attenuation by an Integrated Nature-Based and Engineered Flood Defense System in the Scheldt Estuary (Belgium)"

_jmse, doi:10.3390/jmse8010027_

Round 1

Reviewer 1 Report

In this paper the implementation of culvert flow equations is written in TELEMAC which is able to model the water exchange between the Scheldt estuary and FCAs, and also illustrated the effectiveness in storm surge attenuation of a nature-based and engineered flood defense system in the Scheldt estuary. But following questions should be answered.

The model describes the mathematical formulation for storm surge attenuation for the existing culvert in Scheldt estuary which should be estimated before the construction and surely it happened then what are the innovations there?
The paper does not describe any model formulation for future storm surge.
There are many calibrated culvert parameters have used for Inflow and outflow discharge but are they stay the same for any storm surge? Or they convey different values for different types of storm surge?

Why the Froude scale factor of 15 was chosen in order to use the largest possible dimensions that fitted within the small flume?

In the existing model, almost no water entered the FCA over the overflow dike. If the storm tide is very high then much water will enter in the FCA over the overflow dike. Then what will happen? Does the model contain any specific situation for this case?

The paper said the differences between measured and calculated discharges were small (maximum difference was 6.8 %). But this difference is estimated for just one storm surge if we compare with more samples what will be the percentage as well as the differences?

The paper discussed about 6 types of flow, but there were no details about it.

In the mathematical formulation for energy equation no details were given which term are neglected or not. It’s quite vague. 

Reviewer 2 Report

Smolders et al.

General comments

The manuscript is about modelling storm surge attenuation by an integrated nature-based and engineered flood defence system in the Scheldt estuary (Belgium). Authors present simulation of the storm surge to a hindcast using a 23 TELEMAC-3D model for the Scheldt estuary. An innovative aspect of the study was the simulation of model formulations for calculating flow through culverts of restored marshes while carrying out a physical model study to validate it. The following aspects have to be addressed before its final publication.

Authors have given sufficient introduction. The manuscript can be further generalized if the authors have carried out simulations for flood events with 50 or 100 year return period? The present results may present as a validation to show the capability of the model to a particular estuarine system or specific conditions, for instance, the case of superposition of spring high tide and storm surge. Such results can be important for implementing mitigation plans or adaptation measures.

Details of the validation of the models are not sufficient. Authors have to present the improved validation results of TELEMAC-3D model for the particular FCA with CRT. It also should include an error analysis. Authors have to describe how they maintain not only the linear similarities but also flow similarities of the physical scale modelling part.

Presentation of results has to be further improved with more analysis of the results and error. The authors can estimate the degree of attenuation of storm surge heights if they can carry out the modelling for several storm surge cases. The case of storm surge with 19 year return period seems not sufficient for deriving management decisions as the flood threat would be high in an era of climate change and sea-level rise.

The discussion should include limitations of the model and how the assumptions will have an effect on the results. The discussion also includes a scale-down effect of the estuarine flow and friction due to close boundaries under physical modelling.

Specific comments

Line 47: The word attenuation is self-explanatory no need to include a reduction in brackets.

Line 47: intertidal wetlands can attenuate storm surge instantaneous water levels. It is not required to mention only about peak water levels. It may be possible to use the percentage of attenuation of storm surge peak water levels as a measure of the efficiency of the flood defence mechanism implemented in a particular coastal system.

Line 50-52: very few studies? Do you mean negative sense or you may use a few studies while identifying them with suitable citations?

Line 53-55: for a specific storm surge? Do you suggest that any specific event observed in the past? For generalization, it is appropriate to identify the storm surge with the return period. For instance, storm surge with a 50 or 100 year return period.

Line 57: Cause of the increase in storm surge levels should be identified while citing corresponding citations.

Line 59-61: This storm surge was a combination of a spring tide and strong NW winds? Do you mean that this flood event was a superposition of spring tide and storm surge due to strong NW winds?

Line 62: You have identified the return period for the given location. It is better to mention it from the beginning. Why do you select this particular flood event? It should be rationalized in addition to its cultural coincidence.

Line 72: “so-called flood control areas, FCA”? It may not be appropriate to use so-called. Or else, do you want to suggest that the phrase “flood control areas” is not appropriate?

Line 86: It may not be appropriate to use so-called. Or else, do you want to suggest that the phrase ‘controlled reduced tide’ is not appropriate?

Line 87 and 93: You can use abbreviations (FCA and CRT) after you define them once.

Line 97-98: The validation of the modelling approach should be given under Materials and Methods while citing suitable studies or giving validation results. Identify the FCA used for validation.

Line 97-102: The section is appropriate for the Materials and Methods.

Line 106: Figure 1b? If you identify figure 1b first change 1b-> 1a and 1a -> 1b. Or identify figure 1a first to maintain the order.

Line 109-110: (km 2, Figure 1)?

Line 110:  flood tide enters twice a day? In the beginning, you may mention that the tides of the Scheldt estuary are semi-diurnal.

Line 116: include a coordinate frame in figure 1.

Line 120: Scheldt estuary is a hypersynchronous estuary, where tidal damping by friction is less than the effect of the upstream convergence. Therefore, tides amplify upstream. Identify the correct terminology in your manuscript as much as possible.

Line 146: You already defined CRT.

Line 159: Do the sketch is to a particular scale? If not, give scales with vertical and horizontal exaggerations?

Line 175: Give the resolution of multi-beam measurements of the bathymetry and the Lidar data set.

Line 190-338: Authors can summarize section under 2.3 Model implementation of culvert flow and detailed description on sub-sections therein can be given in an appendix.

Line 342: km 153

Line 426: Give observed inflow and outflow culvert discharges in figure 6 for comparison. Have you again calibrated the model outputs such as inflow and outflow culvert discharges? If so, why?

Line 427: …was used.

Line 429: … were used

Line 428 to 433: Give the validation results and error analysis.

Line 452: It seems that the authors have only focused on maintaining length scale similarities between the estuarine channels and the flume. However, it is very important to maintain similarities such as Froude and Reynold so that it is possible to maintain the similarity between the flow conditions in the estuary and the flume. Sometimes, it is not possible to maintain both Froude and Reynold similarities. Authors can give how they introduce corrections for such situations.

Line 530: slightly underestimate the water level. Calculate the error indexes such as Root Mean Square Error.

Line 540: Does the table give average discharges? Give the temporal variability modelled discharges for each case graphically.

Reviewer 3 Report

Line 62: For readers it might be useful to have an online link to ref [27]: http://www.vliz.be/nl/catalogus?module=ref&refid=283458 or http://resolver.tudelft.nl/uuid:4d028320-7dfa-46a6-ab77-f79bdccce98c or https://waterberichtgeving.rws.nl/water-en-weer/verwachtingen-water/water-en-weerverwachtingen-waternoordzee/stormvloedrapportages/stormvloedverslagen/download:835

Line 64: On one hand….  It is difficult to find the “on the other hand”. Maybe good to start in line 69 with another paragraph (and a capital).

In line 69 you mention “Flemish government”. I think that many international readers do not understand why the Flemish government has something to do with a Belgian problem.

Line 85: I understand the sequence:

Sigmaplan asked for floodplains Floodplains were created Sigmaplan was redesigned Now floodplains are in the process of changing them into tidal marshes.

Is this interpretation correct? I yes, please make text somewhat more clear. If not, a more rigorous editing is needed.

Line 158: you mention in the figure TAW, but this level is explained much later (line 347). I suggest you explain TAW in or below figure 2.

Line 162: “start-of-art” is superfluous. No-one will use an out-of-date model.

Line162/179: You describe many features of the current Scaldis model, but why is this all needed for this specific study (e.g. how relevant is good salinity modelling, how relevant are 5 vertical layers). Or did you simply use this model, because the model was available?

Line 183: [45] is a key-reference for this paper. But it is difficult to find on the WL-website (especially for non-Dutch speaking users). Maybe you can add the online link: http://documentatiecentrum.watlab.be/imis.php?module=ref&refid=261397

Line 230 and some other locations: variable symbols should be in italic. Also it is the habit to write functions not in italic, so:  sin Θ.

Line 244-248: Your explanation of how to determine the value of C56 from Figure 3 is not very clear.

Line 434, section 2.5. Here suddenly a physical model appears in the paper, without any explanation why a physical model is needed for this study. This section needs an introduction. And at the end of section 2.5 I do not see a clue of what was concluded from the physical tests in relation to the numerical model study. It becomes somewhat more clear after reading section 3.2, but that appears much later in the paper

Line 563: I assume this is the highest observed level during the Sinterklaasstorm. In that case it might be good to stress that the depicted levels do not occur at the same moment. With other words, the slope in the line is not the waterl evel gradient.

Line 611: In figure 9 I see that the model does not give correct values downstream of Saeftinghe. But upstream of Saeftinghe it is not bad at all.

(By the way, upstream and downstream may be confusing here. I assume downstream is seaward, because normal river flow is towards the sea. But in case of a storm surge the flood wave streams from sea inland, and then downstream may be interpreted as landward of Saeftinghe).

Line 651: This sentence has no verb.

Line 692: Conclusions. Maybe it is even more convincing to give real numbers. You may mention that due to FCA and CRT the water along the Scheldt between km 110 and 130 are some 10 cm lower than without these systems and landward of km 130 even more, up to 60 cm (for the condition of the Sinterklaasstorm). Also you may conclude that the saltmarsh of Saeftinghe reduces the  water level between Saeftinghe and km 125 for some 10 cm.

====

(although not really a reference to this paper, it might be interesting for the authors to know that Johan van Veen in the  years 1938-1943 suggested to make a number of FCA’s in the Dutch rivers. He also made computations to prove their effectivity, but not having digital computers this was quite difficult. The plans for FCA’s was never implemented, because after 1953 it was decided to close the tidal inlets completely). See http://resolver.tudelft.nl/uuid:e52e8f90-e8dc-436e-84ed-87f0e31a0ef6)
